# Molecular Oscillator Affects Susceptibility of Caterpillars to Insecticides: Studies on the Egyptian Cotton Leaf Worm—*Spodoptera littoralis* (Lepidoptera: Noctuidae)

**DOI:** 10.3390/insects13050488

**Published:** 2022-05-23

**Authors:** Choukri M. Haj Darwich, Marcin M. Chrzanowski, Piotr P. Bernatowicz, Marta A. Polanska, Ewa Joachimiak, Piotr Bebas

**Affiliations:** 1Department of Animal Physiology, Institute of Functional Biology and Ecology, Faculty of Biology, University of Warsaw, 02-096 Warsaw, Poland; choukri.d@biol.uw.edu.pl (C.M.H.D.); p.p.bernatowicz@uw.edu.pl (P.P.B.); ma.polanska@uw.edu.pl (M.A.P.); 2Biology Teaching Laboratory, Faculty’s Independent Centers, Faculty of Biology, University of Warsaw, 02-096 Warsaw, Poland; mm.chrzanowski@uw.edu.pl; 3Laboratory of Cytoskeleton and Cilia Biology, Nencki Institute of Experimental Biology PAS, 02-093 Warsaw, Poland; e.joachimiak@nencki.edu.pl

**Keywords:** molecular oscillator, circadian clock, susceptibility to insecticides, *Spodoptera littoralis*, Lepidoptera

## Abstract

**Simple Summary:**

The oscillator, the core element of the biological clock, is a mechanism operating at the molecular level that is responsible for the generation of rhythms. It is produced by clock genes that code for proteins, regulating expression of its own and executive genes. Oscillator activity manifests itself at the level of the cells, organs, and the whole organism, thus enabling organisms to perform life functions at the most proper time. An example is the intensification of detoxification processes when the body is exposed to xenobiotics. In this study, we bred *Spodoptera littoralis* larvae (an important crop pest) in continuous light (conditions abolishing rhythms in insects by interfering with the clock) and altered the performance of their oscillator using molecular biology tools. We have shown that the susceptibility of *S. littoralis* larvae to insecticides varies daily, being under the control of a molecular oscillator. This oscillator operates in the fat body, midgut, and Malpighian tubules where it is likely to control the activity of detoxifying enzymes. Our results indicate the role of a molecular oscillator in the metabolism of xenobiotics in *S*. *littoralis* larvae. These findings support the rational use of insecticides at specific times of the day.

**Abstract:**

The molecular oscillator is the core of the biological clock and is formed by genes and proteins whose cyclic expression is regulated in the transcriptional-translational feedback loops (TTFLs). Proteins of the TTFLs are regulators of both their own and executive genes involved in the control of many processes in insects (e.g., rhythmic metabolism of xenobiotics, including insecticides). We disrupted the clock operation in *S. littoralis* larvae by injecting the dsRNA of clock genes into their body cavity and culturing the larvae under continuous light. As a result, the daily susceptibility of larvae to insecticides was abolished and the susceptibility itself increased (in most cases). In the fat body, midgut, and Malpighian tubules (the main organs metabolizing xenobiotics) of the larvae treated with injected-dsRNA, the daily activity profiles of enzymes involved in detoxification—cytochrome P450 monooxygenases, Glutathione-S-transferase, and esterase—have changed significantly. The presented results prove the role of the molecular oscillator in the regulation of larvae responses to insecticides and provide grounds for rational use of these compounds (at suitable times of the day), and may indicate clock genes as potential targets of molecular manipulation to produce plant protection compounds based on the RNAi method.

## 1. Introduction

The biological clock has been revealed to have activity on several levels of biological organization from the molecular to organismal level, thus controlling multiple aspects of life [1,2]. The core of the biological clock is formed by a molecular oscillator which is regulated by transcriptional-translational feedback loops (TTFLs) [3,4]. TTFLs comprise the genes and the protein factors they encode, which are commonly referred to as the clock (oscillator) genes and proteins. Among animals, one of the best characterized molecular oscillators is that of the fruit fly, *Drosophila melanogaster*, which consists of two interlocked TTFLs [5,6]. The first TTFL includes positive regulators of gene expression—the CLOCK (CLK) and CYCLE (CYC) proteins, and negative regulators indirectly repressing gene expression—the PERIOD (PER) and TIMELESS (TIM) proteins. CLK and CYC form heterodimers which promote the transcription of *period* (*per*) and *timeless* (*tim)* genes during the solar day. The transcripts are then translated into proteins, PER and TIM, which bind to each other and enter the cell nuclei where they bind to CLK:CYC complexes to suppress their own expression after dusk. This system is reset by light via the cellular photoreceptor CRYPTOCHROME 1 (CRY1), which directs TIM down the degradation pathway (and ultimately also PER), thus leading to CLK-CYC dependent re-initiation of transcription of *per* and *tim* after dawn. In the second loop, *clk* gene transcription is regulated by two factors: positively by PAR-Domain Protein 1 (PDP1) and negatively by VRILLE (VRI). In response, CLK-CYC positively regulates the transcription of the *Pdp1* and *vri* genes. The supporting role of the molecular oscillator system is played by the clockwork orange factor (CWO), which is a transcriptional repressor that competes with CLK-CYC for binding sites in clock-controlled gene regulatory sequences, including the temporary inhibition of its own expression [7].

The molecular oscillator of lepidopterans is structurally similar to the oscillator of the fruit fly. However, it also has its own characteristics. It comprises two CRY proteins (CRY1 and 2) which have immensely distinct physiological functions, while still maintaining the functionality of the TTFL [8,9,10,11]. CRY1 (homologous to *Drosophila*’s CRY1) plays the role of the cellular photoreceptor, while CRY2 (homologous to mammalian cryptochromes) acts as the major repressor of transcription in the first TTFL of the oscillator [12,13]. In this mechanism, CRY2 performs its function by binding to the CLK:CYC heterodimer, thus inhibiting CLK:CYC-dependent transcription, including that of *per*, *tim*, and *cry2* genes. Much less is known about the second TTFL of the molecular oscillator in lepidopterans. Most of the available data comes from transcriptomic analyses, which indicate the expression of *vri* and *pdp1* in certain tissues and also the rhythmic nature of this expression [14,15,16]. As *vri* and *pdp1* genes code for transcription factors, attention is focused not only on their role in the molecular oscillator system itself, but also on the regulation of physiological and developmental processes, both in terms of circadian control and ontogenesis [17,18,19]. Most recently, the expression of the *cwo* gene has also been associated with the lepidopteran response to exogenous factors altering homeostasis [20].

In insects, expression of genes encoding factors acting in TTFLs was found in the cells of several organs, which indicates that a complex multi-oscillator system functions in these animals [21,22,23]. Locally active oscillators regulate tissue- and organ-specific processes, and are often characterized by complete autonomy in receiving information from the environment, processing it, and generating rhythms. Such oscillators have been described in the cuticular epithelial cells, sensory organs, endocrine glands, gonads, and reproductive tracts, as well as in the organs responsible for food intake, excretion, and nutrient storage (gut, Malpighian tubules, and fat body, respectively) which are characterized by a very high level of metabolism [24,25,26,27,28,29,30,31,32]. Moreover, most of these data on the localization and role of locally acting molecular oscillators in controlling organ-specific processes comes from studies done in fruit flies. This is mainly due to the use of molecular tools that allow the production of strains with locally disrupted expression of molecular oscillator components (in tissues or even in specific cells). In the case of other insects, including lepidoptera, data can be grouped into two categories—first, data presenting the rhythmic course of processes in organs, and second, data proving molecular oscillator activity in the cells forming these organs. However, there is little molecular evidence determining the physiological function of these oscillators, i.e., a link between the TTFL of the oscillators and the rhythmic implementation of specific processes. One of the few that has been fairly well-researched in this respect is the oscillator in the male reproductive system [33,34,35,36,37].

A very important aspect of the biological clock’s role in maintaining homeostasis is the control of the response to stresses, including xenobiotics. Insects have a tremendous impact on ecosystems and the economy. However, research on the daily (and circadian) control of xenobiotic metabolism in insects, including insecticides, is still scarce in comparison to studies done on vertebrates [38,39,40,41,42,43]. Fortunately, in recent years, the current data has been successively supplemented with new reports [44]. Particularly valuable are those studies that use methods related to the manipulation of gene expression to prove the control of xenobiotic metabolism by TTFL elements of the molecular oscillator. Lack of key elements of the second TTFL, in *pdp1* and *cyc D. melanogaster* mutants has been shown to affect detoxification pathways by disorganizing rhythmic expression of genes for P450s and specific esterases, leading to changes of sensitivity to permethrin and malathion in such mutants [18]. At the same time, the concatenation of the first TTFL elements (*per* and *tim* genes) with detoxification pathways was not observed. These findings are confirmed by earlier reports indicating the rhythmic activity of uridine 5’-diphosphoglucosyltransferase (UGT) and P450 (determined by indirect measurements of ethoxycoumarin-O-deethylase activity—ECOD), which were disturbed in insects kept under constant light (conditions abolishing functions of molecular oscillator) [45]. The same studies revealed daily changes of fruit fly susceptibility to acute exposure to fipronil (phenylpyrazole), propoxur (carbamate), and malathion. Oscillator control over the expression of one of the P450s (CYP9M9), and clock-dependent resistance to permethrin was established by silencing per gene expression using the RNAi technique in *A*. *aegypti* [46]. These three examples demonstrate the real and direct involvement of the TTFL elements of the oscillator in the regulation of insecticide metabolism.

However, these studies are limited to two species classified under the same order, Diptera, of which only one is of economic importance. During the preparation of the present manuscript, data was published on the influence of a molecular oscillator on the expression of genes encoding digestion and detoxification enzymes in the lepidopteran representative, *Spodoptera litura*. This was the first report in which these issues were addressed specifically for lepidopterans [20]. Our results further expand the current knowledge in this area, and partially fill the gaps in knowledge on the role of the molecular oscillator in the control of lepidopteran susceptibility to insecticides. We have used the Egyptian cotton leaf worm *Spodoptera littoralis*, one of the most destructive pests in vegetables, fruits, and other crops. We disrupted the operation of the clock in the last instar larvae of *S. littoralis*, by keeping them under continuous light followed by determination of susceptibility to commonly used insecticides. Then, using RNAi technique, we knocked down operation of the molecular oscillator in these larvae and analysed the daily activity of xenobiotic metabolizing enzymes in their fat body, midgut, and Malpighian tubules, as well as the daily susceptibility to insecticides.

## 2. Material and Methods

### 2.1. Insect Breeding

Larvae of *S. littoralis* were reared as described previously [47], in cycles of 16 h light:8 h dark (LD) at 25 °C. The start of the dark period was set as Zeitgeber time 12 (Zt 12) and the start of the light phase was set at Zt 20. The 5th (penultimate) instar larvae prepared for molting (identified by head capsule slippage) were separated from the stock culture and observed hourly for the selection of freshly molted individuals of the 6th instar. Only second gate larvae were collected and used (see [48] for details); in specific experiments, freshly molted larvae were transferred into constant light (LL). The experiments were done on 2-day-old larvae of the 6th instar. All tests were carried out with respect for the animals, and the number of insects used for the analysis was reduced to the necessary minimum. Before all procedures, the larvae were anesthetized in a CO_2_ atmosphere; in addition, insect dissections were performed in ice-cold buffer.

### 2.2. Insect Treatment with Insecticides

Fipronil, deltamethrin, malathion, propoxur, acetamiprid, and imidacloprid were obtained from Greyhound (Greyhound Chromatography and Allied Chemicals Ltd., Birkenhead, UK) and dissolved in acetone—ROTISOLV^®^ GC Ultra Grade (Carl Roth GmbH + Co. KG, Karlsruhe, Germany). Insects were treated with insecticides as described previously for *D*. *melanogaster* [45] with slight modifications (Figure 1A). Two hours prior to exposing caterpillars to insecticide at each time point, the inner walls of 50 mL glass vials with rolled rim (height: 100 mm) (Carl Roth GmbH + Co. KG) were coated with 500 μL of insecticide or of acetone (control). The insecticide concentrations used were as follows: 0.5–250 μg/mL fipronil, 0.1–100 μg/mL deltamethrin, 0.5–120 μg/mL malathion, 0.1–300 μg/mL propoxur, 0.1–350 μg/mL acetamiprid, and 0.1–500 μg/mL imidacloprid. The vials were rolled using a hot dog roller (GGM Gastro International GmbH, Ochtrup, Germany) and warmed to 35 °C until the acetone was completely evaporated. Larvae were individually placed into the insecticide- or acetone-coated vials at Zt (or Ct for insects under LL) 0, 4, 8, 12, 16, 20, and 24 (i.e., repeated Zt/Ct 0 of the 3rd day). After 1 h, the caterpillars were placed in Petri dishes with vents (145 × 20 mm) (Greiner Bio-One GmbH, Frickenhausen, Germany) with a small cube of fresh diet. The same procedure was applied to larvae which, apart from the treatment with insecticides at different time points of the day, were also injected once, i.e., at Zt 0 with dsRNAs of the selected biological clock genes (see below) (Figure 1B). Larvae mortality was checked 48 h after treatment. The concentration of pesticide that was lethal to 50% of tested larvae (LC_50_) was calculated using probit analysis using LeOra Software POLO-Plus [49]. Tests were performed on 10–15 larvae to determine LC_50_ values for each time point and each treatment was performed in three replicates.

### 2.3. Insect Treatment with dsRNA

To disrupt TTFLs of the molecular oscillator, the larvae were injected with 800 pmol of *period* (*per*), *timeless* (*tim*), *cryptochrome 2* (*cry2*), *cycle* (*cyc*), *vrille* (*vri*), *par-domain protein 1* (*pdp1*), or plant (*ribulose-1,5-bisphosphate carboxylase small subunit*—*SSU* from the cultivated tobacco, *Nicotiana tabacum*, which served as a control) dsRNA in 4 μL of transcription buffer. Larvae injected with 4 μL of transcription buffer and untreated larvae (intacts) were used as additional controls. The solutions were injected by inserting a sharpened glass capillary (1–5 μL micropipettes, BLAUBRAND^®^; Brand GmbH + CO. KG, Wertheim, Germany) into the body cavity through one proleg of the second pair. Injections were started at Zt 0 and given every 4 h until the beginning of the next day (Zt 0 of the 3rd day of the last instar) (Figure 1C). Ten to twelve individuals from each dsRNA-treated and control groups were dissected every 4 h in lepidopterans buffer. The collected fat body, midgut, and Malpighian tubules were placed in 1.5 mL centrifuge tubes and frozen in liquid nitrogen. Prior to freezing, the intestine was cut lengthwise to flush the contents using dissection buffer. Organs from one individual were placed in a tube and separate from others (no organs were pooled from different individuals) at each given time point. The buffer introduced into the test tubes together with the organs was maximally aspirated with a pipette. These organs were stored at –80 °C until RNA extraction and measurements of the activity of xenobiotic metabolizing enzymes were performed. Before starting the RNA extraction procedure and enzyme activity analyses, the samples were placed on ice and an appropriate buffer was added in which the tissues were homogenized and then processed according to the procedures described below. All studies on the larvae of experimental and control groups were performed in triplicate.

### 2.4. dsRNA Synthesis

The dsRNA of *per*, *tim*, *cry*2, *cyc*, *vri*, *pdp1*, and *SSU* (plant) was produced as described previously [34,50]. Briefly, Malpighian tubules were used for total RNA extraction with Renozol RNA (GenoPlast Biochemicals, Rokocin, Poland) according to the manufacturer’s instructions, followed by treatment with DNase I–RNase-free (New England BioLabs—NEB Inc., Ipswich, MA, USA). The first strand from the total RNA was synthesized with AMV Reverse Transcriptase (NEB Inc.) and Random Primer Mix (NEB Inc.). Fragments of sequences were amplified by PCR, using Maximo Taq DNA Polymerase (GeneOn GmbH, Ludwigshafen, Germany) and specific sense and antisense primers (Appendix A); fragments of both *cyc* and *pdp1* genes were cloned (see Appendix A). DNA templates were transcribed in vitro in two separate reactions with SP6 and T7 RNA polymerases (NEB Inc.). The sense and antisense RNA probes were designed to produce ~300 bp fragments of selected gene sequences. The efficiencies of in vitro transcriptions were verified on agarose gels. Using the same strategy, control non-specific (plant) dsRNA was synthesized using the gene sequence for *SSU* from *N. tabacum* [34]. The synthesized RNAs were kept at −80 °C.

### 2.5. Enzymatic Assays

The next step was to abolish the oscillator’s operation in larvae using the RNAi technique followed by an analysis of detoxifying enzyme (ECOD, GST, and esterase) activities in the fat body, midgut, and Malpighian tubules—the organs mainly responsible for the metabolism of xenobiotics. The dsRNAs produced by in vitro transcription, based on the coding fragments of selected *S. littoralis* clock genes—*per*, *tim*, *cry2*, *cyc*, and *pdp1* (and *SSU* of *N. tabacum*), were injected into the body cavity of the larvae. Then the mRNA level of targeted genes were determined in the organs as described in the Appendix A. Results obtained for larvae injected with the dsRNA of clock genes were then compared against data we received for untreated larvae, injected with buffer used for in vitro transcription, and injected with non-specific plant dsRNA. Since the disruption of clock gene expression using dsRNA to abolish oscillator function is only a tool to determine its potential role in regulating insecticide metabolism, these data are also provided in the Appendix A (Appendix A).

#### 2.5.1. 7-Ethoxycoumarin-O-deethylase (ECOD) Activity

The total cytochrome P450 monooxygenase activity was measured as described previously [51,52], with modifications made for testing in insects [53,54]. These procedures are based on measuring P450 dependent 7-ethoxycoumarin-O-dealkylation (ECOD) activity, with 7-ethoxycoumarin (7-EC, 7-Ethoxy-1-benzopyran-2-one) as a substrate. Collected organs were homogenized using Micro Potter Elvejham homogenizer (Ace Glass Inc., Vineland, NJ, USA) in 0.05 M sodium phosphate buffered saline (PBS–Na) pH 7.4 containing 1 mM EDTA, 0.4 mM PMSF, and 0.1 mM DTT. Midguts were homogenized in 1 mL buffer, while the fat body and Malpighian tubules were homogenized in 200 µL buffer. Then, the homogenates were centrifuged at 10,000× *g* and the pellet was discarded. The solution was centrifuged a second time at 100,000× *g* and the supernatant was discarded. The pellet obtained with the microsomal fraction was suspended in the buffer previously used to homogenize tissues and protein concentration was determined. The reaction mixture containing the microsomal fraction (250 µg proteins), 0.4 mM 7-ethoxy-2H-1-benzopyran-2-one, and 1 mM NADPH (both from Cayman Chemical, Ann Arbor, MI, USA) in a final volume of 0.5 mL PBS was placed in 1.5 mL tubes and incubated for 30 min at 30 °C. For triplicate measurements, 150 µL of the reaction mixture was applied to three wells of a Nunc™ F96 MicroWell™ Black Polystyrene Plate (Nunc, Roskilde, Denmark). The fluorescence was measured at 380 nm excitation and 450 nm emission using a VICTOR X3 Multilabel Plate Reader (PerkinElmer, Waltham, MA, USA). The amount of 7-hydroxycoumarin formed by ECOD was determined with a standard curve established from known amounts of 7-hydroxycoumarin. The ECOD activity was expressed as the nmol of product formed per minute per mg of protein.

#### 2.5.2. Glutathione-S-transferase (GST) Activity

The GST activity was determined according to procedures described previously [55,56]. Organs were disintegrated by sonication using SONOPULS HD 2070 ultrasonic homogenizer (BANDELIN electronic GmbH & Co. KG, Berlin, Germany) in 0.1 M potassium phosphate buffered saline (PBS-K) pH 7.2 with 0.3% Triton X-100 (the intestine from one larva was homogenized in 1 mL buffer, while the fat body and Malpighian tubules in 200 µL buffer), followed by two rounds of centrifugation: first for 5 min at 3000× *g* then 20 min at 10,000× *g*. The pellets were discarded and the concentration of proteins in the second supernatant was determined. Twenty µL of diluted supernatant (1:5 with PBS-K) was applied to the well of a Greiner UV-Star^®^ 96 well plate (Greiner Bio-One GmbH) containing 170 µL of PBS-K and 20 µL 5 mM glutathione (GSH), followed by the addition of 10 µL of 1.2 mM 1- chloro-2,4-dinitrobenzidine (CDNB). Enzyme activity was measured using the TECAN Infinite M200 microplate reader (Tecan Trading AG, Männedorf, Switzerland) at 340 nm and 30 °C for 10 min using the kinetic mode. GST activity was calculated using the Beer–Lambert Law where ε_340_ = 9.6 mM^–1^ · cm^–1^. The obtained results were presented as the nmol of glutathione conjugate formed per minute per mg of protein.

#### 2.5.3. Esterase Activity

Esterase activity was measured according to the procedure described previously and used for studies of lepidopterans, including *S*. *littoralis* in the context of insecticide metabolism [57,58], with slight modifications. Briefly, organs were disintegrated by sonication using an ultrasonic homogenizer (see above) in 0.05 M PBS-K pH 6.8 with 0.3% Triton X-100 (the intestine from one larva was homogenized in 1 mL buffer, while the fat body and Malpighian tubules in 200 µL buffer), then centrifuged at 10,000× *g* and protein concentration was determined. Equal volumes (120 µL) of supernatant and 0.1 mM α-naphthyl acetate on PBS-K were mixed in a 0.5 mL test tube and incubated for 30 min at 35 °C. For the controls, the protein extracts from the organs were replaced with an equal volume of PBS-K. Then, 120 µL of 0.3% Fast Blue RR Salt (Otto Chemie Pvt Ltd., Mumbai, India) dissolved in double distilled water containing 5% SDS was added to stop the reaction. Ten minutes after development of color, 100 µL of reaction mixture was pipetted into three consecutive wells of the 96 well polystyrene microplate (Greiner Bio-One GmbH) to measure esterase activity in each organ in triplicate. The plates were read at 595 nm in VICTOR X_3_ Multilabel Plate Reader and the concentration of products was determined from standard curves of α-naphthol. The esterase activity was expressed as the nmol of product formed per minute per mg of protein.

As all enzymatic assays required determination of the total protein level in the samples, a commercial BCA Protein Assay Kit was used for this purpose in accordance with the manufacturer’s instructions (Cell Signaling Technology Inc., Danvers, MA, USA). This procedure included the standard curve determination on quantitatively standardized bovine serum albumin solutions for estimation of protein concentration.

## 3. Results

### 3.1. Exposure to Continuous Lighting Changes the Larvae’s Susceptibility to Insecticides

The average LC50 for fipronil changed over 24 h for LD larvae, with the highest values observed in the evening (Zt 8), followed by a drop to the lowest value at the time of transition from night to day (Zt 20), and then an increase during the day (Figure 1A). It is worth noting that a significant difference between the maximum and minimum LC50 was nearly three and a half times. Furthermore, a cosinor analysis of the data indicates that observed daily changes in larvae sensitivity to fipronil are rhythmic (Appendix A—in the caption for this table, a brief description of the cosinor method used here and literature references to it are provided). In LL insects, no rhythmic susceptibility to fipronil during the 24 h was demonstrated, as the average LC50 remained at a comparable level at all tested time points of the day. In addition, the LC50 values for this insecticide in LL insects were significantly lower almost in all time points over the day if compared to the LC50 in LD insects (with the exception of the Zt/Ct 20). These differences were smallest in the middle of the night (Zt/Ct 16) and the largest in the evening (Zt/Ct 8), of about two and four times, respectively.

For deltamethrin, the average LC50 values for the LD group also changed cyclically during the day, with a significant increase during the night hours, reaching the maximum value at the time of transition from night to day (Zt 20) followed by a decrease in the morning and reaching a minimum in the middle of the day (Zt 4) (Figure 1B). It should be noted that in the case of deltamethrin, the difference between the highest and the lowest LC50 values recorded during 24 h was the most significant (when comparing the effects that are caused by other tested insecticides), and this difference was greater than thirteen times. Moreover, the analysis of changes in the daily susceptibility of larvae to this insecticide shows that it is rhythmic (Appendix A). Such cyclic changes were not found in LL insects treated with deltamethrin; the average LC50 values for deltamethrin did not differ significantly in this case. On the other hand, the comparison of the LL and LD groups showed a clearly lower LC50 for deltamethrin in insects from LL conditions in the middle and at the end of the night, whereas it was lower by half at Zt/Ct 16, and nearly four times lower at Zt/Ct 20.

Another rhythm profiles the diurnal changes in malathion sensitivity in LD larvae (Appendix A). A significantly higher average LC50, and therefore, a higher resistance to it, was observed at the time of transition from the day to the night (Zt 12) and at midnight (Zt 16) (Figure 1C). The lowest LC50 was recorded in larvae treated in the first half of the day (Zt 0/24) wherein the difference between the minimum and maximum LC50 was nearly three-fold. It was obvious that switching on the light in insect breeding (beginning of the day) was accompanied by a decrease in the average LC50 for malathion, while switching off the light (beginning of night) was associated with an increase in LC50. For LL insects, such daily changes were not observed, and the average LC50 values for malathion remained similar throughout the day and were statistically different at each time point from the values of LC50 determined for the LD insects. The smallest of these differences was almost three times (Zt/Ct 0/24) and the highest was more than ten times (Zt/Ct 12).

The highest average LC50 values for propoxur administered to LD larvae were also recorded at night (Figure 1D). LC50 was lowest in the morning (Zt 0/24) and increased during the day, reaching a maximum at the beginning of the night (Zt 12). The LC50 then decreased, remaining at a comparable level for the rest of the night, and dropped at the beginning of the day in treated larvae. The increase from the minimal to the maximal LC50 here was less than two and a half times. These changes, nevertheless, have the hallmarks of a rhythm, as demonstrated by cosinor analysis (Appendix A). We did not observe such a cycle in the larvae kept in LL conditions. Moreover, the LC50 value for propoxur in LL insects was significantly lower than in LD insects, with a maximum eleven-fold difference (Zt/Ct 12) and a minimum difference of about three and a half times (Zt/Ct 4).

Daily differences in the average LC50 values for acetamiprid administered to LD group were significant. The highest LC50 values for this insecticide were recorded in the larvae treated in the evening (Zt 8), while the lowest values were seen when larvae were treated at the end of the night (Zt 20); the difference between these values was two and a half times (Figure 1E). Although the amplitude of these changes was not very large, the process still showed hallmarks of a rhythm as shown by cosinor analysis (Appendix A). Such cyclic changes in LC50 for acetamiprid were not found in LL larvae. The differences in susceptibility expressed as the LC50 between insects from LL and LD conditions were only concerned with the late day and early night period (Zt/Ct 8–12). At these time points, the LC50 values for LD larvae treated with acetamiprid were 50 to 60% higher than for LL larvae.

The difference between the minimum and maximum average LC50 for imidacloprid applied in larvae was nearly two times (between Zt 8 and Zt 12, respectively), and as with other insecticides tested, the larvae showed diurnal differences in susceptibility to this compound (Figure 1F). Furthermore, the changes showed signs of a biological rhythm (Appendix A). Such a cycle was not observed in larvae treated with imidacloprid and kept under LL conditions. However, here, a clear trend was observed, indicating that the average LC50 for this insecticide in larvae treated and housed in LL exceeded (at Zt/Ct 12 and 16 by about 50%) or was close to exceeding (at Zt/Ct 0/24 and 20, despite no statistically significant differences) the average LC50 values found at a given time point for LD larvae.

### 3.2. Knocking Out Clock Genes Expression through RNAi Affects the Detoxification Enzymes Activity in the Fat Body, Midgut, and Malpighian Tubules

#### 3.2.1. Fat Body

It was observed in the fat body of larvae from the control groups that the expression of all clock genes studied changed every 24 h (Appendix A). The profiles of these daily oscillations are specific for each gene, with an easily distinguishable peak and trough whose values were statistically different. Furthermore, cosinor analysis showed that the expression of these genes in insects from control groups had features of the rhythm (Appendix A). The dsRNAs of particular clock genes injected into the insects caused a radical reduction in the level of fat body transcripts of these genes and completely abolished the rhythmic nature of the process.

In the fat body of larvae from all control groups, the activity of ECOD changed cyclically throughout the 24 h (Figure 2A). The mean level of its activity increased during the day, reaching a maximum at the beginning of night (Zt 12) and then dropping sharply (nearly three times) until the end of the night (Zt 20), and rising again in the morning. These changes have the hallmarks of a rhythm as indicated by cosinor analysis (Appendix A). In order to maintain clarity of data, on the graphs (in this and the following figures—Figure 2, Figure 3 and Figure 4), we show only the results of statistical analysis obtained for the control larvae injected with non-specific plant dsRNA. In animals that were injected with the dsRNA of the *per*, *cyc*, and *pdp1* genes, the mean level of fat body ECOD activity at each time point was very low throughout the entire day and significantly lower at most time points than in the tissues of animals from the control group (Figure 2A,D,E). There were also no features indicating its rhythmic character during the 24 h in insects treated with these dsRNAs (Appendix A). The situation was completely different in the case of insects treated with *tim* and *cry2* dsRNA, in which the profile of the daily ECOD activity in fat body not only changed analogously to that of control insects (Figure 2B,C), but these changes were also rhythmic (Appendix A) with a maximum and a minimum at the beginning (Zt 12) and at the end of the night (Zt 20), respectively (and this difference was almost two and a half times).

The activity of GST in the fat body changed cyclically during the 24 h in larvae from the control groups (Figure 2F). The average GST activity at all time points of the day remained at a comparably low level, and then rose sharply at the beginning of the night (between Zt 12 and Zt 16) and peaked at the end of the night (Zt 20; increasing more than twelve times), followed by its immediate decline in the morning (Zt 24/0). Furthermore, these changes were rhythmic as shown by cosinor analysis (Appendix A). In the larvae treated with *per* dsRNA, GST activity was not rhythmic. It remained at a comparable level throughout the 24 h and was substantially higher (up to eleven times) during the daytime when compared to the levels observed at the same daytime points in the control (Figure 2F). These mean values of GST activity in all time points for *per* dsRNA-treated larvae were close to the highest activity recorded for the larvae of the control group during the second half of the night (between Zt 16 and Zt 20). In larvae injected with *tim* dsRNA, the level of GST activity changed rhythmically (Appendix A), keeping the shape of the 24 h rhythm almost identical to the control, with activity at a minimum in the evening (Zt 8) and a maximum in the second half of the night (between Zt 16 and Zt 20), in which the increase in GST activity was more than nineteen-fold (Figure 2G). In insects treated with *cry2* dsRNA, cyclic changes in GST activity were also observed with a morning minimum (Zt 0/24) and an end-of-night maximum (Zt 20) (Figure 2H). These changes had the characteristics of a 24 h biological rhythm (Appendix A). In this case, however, the amplitude of the rhythm was lower than in the control. Moreover, even with a rhythm profile similar to that observed in insects from the control group, the average values of GST activity in *cry2* dsRNA larvae recorded during daytime (between Zt 0/24 to Zt 12) were about five to seven times higher at the same time points when compared with insects from the control groups. In insects injected with both *cyc* and *pdp1* dsRNA, the activity of GST was very low throughout the day and remained at the same level as in the daytime hours of insects from the control groups (Figure 2I,J). The cosinor analysis of the data did not show that it varied rhythmically (Appendix A).

Analyses of esterase activity in the fat body of larvae from control groups showed that it changed cyclically throughout the 24 h, reaching a minimum during the daytime (Zt 4) with a gradual increasing to a maximum (more than eight-fold) at the end of the night (Zt 20) (Figure 2K). Nevertheless, these changes do not have the features of the rhythm as shown by cosinor analysis (Appendix A). In larvae treated with *per*, *tim,* and *cry2* dsRNA, the mean esterase activity was markedly elevated when compared with that of the control group (at each time point of the 24 h daily cycle) (Figure 5K–M). These differences were maximal in the middle of the day (Zt 4) and correspond to an almost ten-fold increase in activity in *per* dsRNA insects, nearly eight-fold in *tim* dsRNA insects, and over ten-fold in *cry2* dsRNA insects. Moreover, the activity of GST in the fat body did not change in a rhythmic fashion in either of these groups (Appendix A), although there was a significant increase in esterase activity following administration of *per* and *tim* dsRNA to insects within 24 h (in both cases, the increase was one and a half times). Conversely, when the larvae were treated with *cyc* and *pdp1*, the mean level of esterase activity in the fat body was lowered at most time points of the 24 h diurnal cycle when compared with the values found for the control insects (Figure 2N,O). In this case, we also did not observe that this activity had the specificity of a biological rhythm (Appendix A).

#### 3.2.2. Midgut

In the midgut of larvae from the control groups, the expression of all clock genes studied changed over a 24 h period (Appendix A). Profiles of these daily patterns of expression were specific for each gene, with an easily distinguished peak and trough whose values were statistically different. In addition, cosinor analysis showed that except for the *per* gene, the expression of all analyzed clock genes in the midguts of larvae from the control groups had features of the rhythm (Appendix A). Moreover, dsRNAs of *per*, *cry2*, *cyc*, and *pdp1* genes injected into the larvae caused a radical reduction in the level of their transcripts, and in the case of the latter three, completely abolished the rhythmic nature of the process (Appendix A and Appendix A). The administration of *tim* dsRNA to insects, which primarily changed the daily expression profile of *tim*, disrupted the rhythm of its expression but reduced transcription to a lesser extent.

The average activity of ECOD changed over the day in the midgut tissues of larvae from the control groups, but the differences between the maxima and the minima did not exceed the value of one and a half (Figure 3A). Moreover, as demonstrated by the cosinor analysis, these changes did not show the oscillation characteristic for the rhythm (Appendix A). In larvae treated with *per*, *cyc*, and *pdp1* dsRNAs, we found a clear and significant reduction in the average level of ECOD activity at each time point within 24 h. The greatest decrease of activity in the larvae treated with *per* dsRNA versus control larvae was recorded at midday (Zt 4), decreasing by nearly nine times, with *cyc* dsRNA by more than thirteen times at midday (Zt 4), and with *pdp1* dsRNA by more than twelve times in the evening (Zt 8) (Figure 3A,D,E). Nevertheless, as in the control group, the values of the average ECOD activity in the midgut of these dsRNA-treated larvae at subsequent time points of the day remained at a comparable level, indicating that the process was not rhythmic (Appendix A). In turn, administration of *tim* dsRNA to larvae did not affect the level of midgut ECOD, while the administration of *cry2* dsRNAs increased (a maximum of approximately two and a half times at Zt 0/24) the level of midgut ECOD activity. Additionally, in both cases, the activity remained at a comparable level for 24 h (Figure 3B,C), which was not rhythmic as shown by cosinor analysis (Appendix A).

The activity of GST in the midgut of larvae from control groups changed over a 24 h cycle (Figure 3F). Its average values remained at a comparatively low level from morning until the evening (between ZT 0/24 and ZT 8) which was when activity started to increase (from Zt 8), reaching the highest values at the end of the night (Zt 20) (increasing by about four times), and then dropping rapidly to the level observed during the day (Zt 24/0). These changes have the features of a rhythm, as shown by cosinor analysis (Appendix A). In the larvae from the *per* dsRNA group, we observed an increase in the average GST activity in the midgut at almost all time points as compared to the control (the largest difference, amounting to four times, was noted in Zt 4) (Figure 3F). The activity was lowest in the morning (Zt 0/24) and highest at the end of the night (Zt 20), but the increase was much milder (if compared with the profile of control) and the difference between these minimum and maximum values was about one and the half times. These 24 h changes also have the hallmarks of rhythm, but are, nevertheless, of relatively low robustness, as shown by the cosinor analysis (Appendix A). In the case of insects treated with *tim* and *cry2* dsRNA, the profile showing the diurnal changes in GST activity in the midgut was very similar to that shown in the control larvae, with a pronounced minimum during daytime (Zt 4 for *tim* dsRNA larvae and Zt 8 for *cry2* dsRNA larvae) and a maximum at the end of the night (Zt 20) (Figure 3G,H). In the *tim* dsRNA larvae, this increase was nearly by three and a half times, and in the case of *cry2* dsRNA, it was nearly by two and a half times. Moreover, in the midgut of the *cry2* dsRNA larvae, the mean GST activity at almost every time point was significantly higher than in the control (up to three times that of the control at Zt 4). As shown by the cosinor analyses for the larvae of both groups, the daily changes in GST activity were rhythmic (Appendix A). In the larvae treated with the *cyc* and *pdp1* dsRNA, we found a very low level of GST activity throughout the 24 h. Therefore, in relation to the values from the night hours in the control (when the GST activity was highest), we observed a decrease in activity in larvae treated with dsRNA, by nearly five times (at Zt 20 in *cyc* dsRNA larvae), and even more than five and a half times (at Zt 20 in *pdp1* dsRNA larvae). We also observed an obvious flattening of the daily GST activity profile in the midgut in the larvae from these groups, which is reflected in the lack of rhythm of this process (Appendix A).

The average activity of esterase in the midgut of the larvae in the control groups changed in the 24 h mode, showing a clear minimum during the daytime (between Zt 4 and Zt 8), a successive increase in the evening (Zt 8) and throughout the night, until the first hours of the next day (Zt 24/0–0/24), at which point it reaches its maximum and then drops rapidly (Figure 3K). This increase from minimum to maximum was nearly three-fold. The daily changes in the activity of esterase in the midgut also showed clear signs of the rhythm, which is confirmed by the cosinor analysis (Appendix A). In the case of the knockout of the expression of the three clock genes *per*, *tim*, and *cry2*, the midgut esterase activity of these larvae was significantly higher (at almost every time point) when compared to insects from the control group (Figure 3K–M). In the case of larvae from the group of *per* dsRNA, this maximum difference was nearly three and a half times; more than four and a half times in *tim* dsRNA, and nearly four times in *cyr2* dsRNA for each case recorded in the evening (Zt 8). The opposite situation was found in larvae treated with *cyc* and *pdp1* dsRNAs when compared to the control, because in both cases, a decrease in the mean activity of esterase in midgut at almost every time point of the day was observed (Figure 3N,O). The greatest differences were observed in the morning (nearly eight and a half times in Zt 0/24 in *cyc* dsRNA larvae, and nearly six times in *pdp1* dsRNA larvae). Moreover, in each group of larvae treated with clock gene dsRNA, there was a flattening of the 24 h profile of esterase activity in the midgut, which made the process lose its rhythmic character (Appendix A). Such lack of rhythm was observed even if the differences between the values recorded for subsequent time points were statistically significant (as was observed in larvae treated with *per*, *cry2*, and *cyc* dsRNAs).

#### 3.2.3. Malpighian Tubules

Another organ we examined were the Malpighian tubules, which were characterized by the rhythmic expression of all biological clock genes that we have analyzed (Appendix A). The profile of these expressions varied for different genes, with the maximum and minimum values observed at different time points of the 24 h cycle. Moreover, the cosinor analysis showed that the expression of each of the studied genes was significantly rhythmic (Appendix A). After administration of dsRNAs *per*, *cry2*, *cyc*, and *pdp1*, their expression dropped to low values, which led to a more or less pronounced flattening of their profiles and loss of the rhythm of expression of these genes in the treated insects (Appendix A). Another effect was observed after administration of *tim* dsRNA to insects, wherein the rhythm disappeared but the expression profile itself was largely maintained with a pronounced maximum at the same time point of the day as in the insects from the control groups.

The average activity of ECOD in the Malpighian tubules of larvae from all control groups changed during the 24 h cycle, reaching minimum values in the first half of the day (Zt 0/24) and increasing until the middle of the night, followed by a rapid decrease to a minimum by the next morning (Zt 20–24/0) (Figure 4A). The difference between the minimum and the maximum was close to six and a half times. These distinct diurnal changes were also reflected in the rhythmic nature of the process, as indicated by the cosinor analysis (Appendix A). Treatment of larvae with *per*, *cyc*, and *pdp1* dsRNA dramatically reduced the average level of ECOD activity in Malpighian tubules to the lowest values observed in the control group (and in the case of *cyc* dsRNA-treated larvae, had even lower values which remained at the limit of detection), and this state is maintained throughout the whole day (Figure 4A,D,E). The greatest differences in ECOD activity between the larvae from both the control and treatment groups (observed in the middle of the night–Zt 16) were about nine times for *per* dsRNA larvae, more than forty times for *cyc* dsRNA larvae, and a little more than four times for *pdp1* dsRNA larvae. The knockout of the expression of these clock genes by administration to the larvae of dsRNA was also reflected in the complete loss of the rhythm of ECOD activity in Malpighian tubules (Appendix A). In contrast, larvae treated with *tim* and *cry2* dsRNAs had no significant effect on the average daily changes in the ECOD activity in Malpighian tubules (Figure 4B,C). The profile of changes in its activity was almost identical in both cases as in the control group, with pronounced minima in the first half of the day (Zt 0/24—4) and maximum levels in the middle of the night (Zt 16). Ultimately, the ECOD activity in *tim* and *cry* dsRNA larvae had the features of a robust rhythm (Appendix A).

The average activity of GST in Malpighian tubules in the larvae from the control groups changed cyclically within 24 h, remaining at a comparable (quite low) level in the morning and in the first half of the day (Zt 0/24–4), then dropping to a minimum in the evening (Zt 8), and rapidly increasing to the highest values in the first hours of the night (Zt 12–16) (Figure 4F). In the second half of the night, we observed a downward trend in this activity which lasted until the morning (Zt 24/0). The difference between the average minimum and maximum values of GST activity (detected at Zt 8 and Zt 16, respectively) was almost three-fold. Moreover, the cosinor analysis proved that the described changes in insects from the control groups have the features of a rhythm (Appendix A). Administration of *per* and *cry2* dsRNA to larvae had a similar effect. Both treatments resulted in a nearly constant level of GST activity in the Malpighian tubules during the day (Figure 4F,H). Moreover, the average GST activities at almost every time point in these larvae were comparable to the highest activity of that recorded in the control larvae (at Zt 16). This also resulted in a loss of the GST activity rhythm in the Malpighian tubules of these larvae (Appendix A). In turn, the average level of GST activity in the Malpighian tubules of larvae injected with *tim* dsRNA was not significantly affected (Figure 4G). The profile of changes in these insects was very similar to that observed in the control group with the evening minimum (Zt 8) and the maximum extended to the night hours (between Zt 12 and 16), and the process had a rhythmic character (Appendix A). A very low GST activity in Malpighian tubules, maintained at a comparable level during the 24 h cycle, was found in insects injected with *cyc* and *pdp1* with dsRNA (Figure 4I,J). The mean value of this activity in insects from both experimental groups at almost every time point of the day was significantly lower than in the control larvae. This difference was greatest, about ten times, at midnight (Zt 16) in *cyc* dsRNA insects, and about five times at the beginning of the night (Zt 12) in *pdp1* dsRNA insects. Moreover, the flattening of the daily GST activity profile observed in both cases resulted in the process being non-rhythmic (Appendix A).

The average esterase activity in the Malpighian tubules varied over the 24 h cycle in the larvae of the control groups (Figure 4K). The low value remained during the day until the evening (Zt 0/24–8), when it started to increase to its maximum values in the second half of the night (Zt 16–20), and then dropped again to the values characteristic for daytime (Zt 24/0). The maximum differences observed (between the peak and the trough) indicate a nearly twelve-fold increase in esterase activity between day and night in Malpighian tubules. Moreover, different values determining changes in this activity through the 24 h were arranged in a profile that the process after the cosinor analysis can be considered as rhythmic (Appendix A). Administering dsRNAs of genes encoding negative transcriptional regulators of the molecular oscillator, *per*, *tim* and *cry2*, led to two changes in the average esterase activity in Malpighian tubules: (1) a slight increase of this activity in the course of the day (since application of the dsRNAs), and (2) a significantly higher activity in the daytime when compared to the activity in the control (in *per*, *tim*, and *cry2* insects were the highest, i.e., almost eight-fold, seven-fold and twelve-fold differences in relation to control, respectively, were found at Zt 8) (Figure 4K–M). As a result, insects treated with *per*, *tim*, and *cry2* dsRNAs lost the rhythm of esterase activity in Malpighian tubules (Appendix A). When *cyc* and *pdp1* dsRNA were administered to the insects, the average level of esterase activity in Malpighian tubules was kept at a constant, very low level throughout the day and night (Figure 4N,O). Compared to the control, when esterase activity was maximal (at Zt 16), this difference was more than twenty-fold for *cyc* dsRNA insects and more than five-fold for *pdp1* dsRNA insects. Moreover, such a flattening of the daily esterase activity profile after administration of dsRNA to larvae resulted in the disappearance of the rhythm observed in this case in the control (Appendix A).

The daily activity of the tested detoxification enzymes in all examined organs in insects treated with *pdp1* and *cyc* dsRNA was radically reduced and non-rhythmic. Such a regular pattern of responses to specific dsRNAs was not found in the case of other enzymes. The disruption of expression of both *per* and *cry2* appeared to have opposite effects on GST and esterase activity than *pdp1* and *cyc*. After knockdown of the *per* and *cry2* genes, GST and esterase activity increased in organs, but at some time points of the day, its value remained comparable to the control values. On the other hand, in the case of ECOD, it was difficult to find a pattern between the treatment of insects with dsRNAs and the enzymatic activity in their fat body, midgut, and Malpighian tubules. Knock down of *tim* and *cry2* genes expression using dsRNA did not seem to affect the rhythmic activity of ECOD in fat body and Malpighian tubules, while dsRNA of the *per* gene abolished its rhythm and lowered its level in all of the examined organs. However, in the midgut, the dsRNA of the *tim* gene did not affect the activity of ECOD, and the *cry2* dsRNA significantly increased it.

### 3.3. Knocking out of the Clock Genes Expression through RNAi Changes the Larvae’s Susceptibility to Insecticides

The average LC50 from fipronil exposure changed over 24 h, with its highest values observed in the evening (Zt 8) and the lowest value at the end of the night (Zt 20), and the difference between the two was more than three-fold (Figure 5A). As a result, the cosinor analysis showed that the changes of susceptibility to fipronil in control insects had signs of a rhythm (Appendix A). The administration of *per* dsRNA to larvae significantly reduced the average LC50 for fipronil at almost every time point of the day to a value close to the lowest value described for insects from the control group, ultimately abolishing the fipronil susceptibility rhythm (expressed as changes in the LC50) in such treated larvae (Figure 5A). On the other hand, administration of *tim* and *cry2* dsRNAs did not substantially affect the daily changes in the average LC50 value for fipronil in such treated larvae. In this case, as in the control, the highest LC50 value was recorded during the daytime (at Zt 8 in *tim* dsRNA insects and at Zt 4 in *cry2* dsRNA insects), which was more than three and a half times higher than the lowest observed at the end of the night (Zt 20) (Figure 5A). As resulted in both groups, we found that a biological rhythm in susceptibility to fipronil was maintained, with characteristics similar to the rhythm described for the control (Appendix A). Injection of both *cyc* and *pdp1* dsRNA had a very significant impact on the diurnal susceptibility profile of the larvae to fipronil. The mean LC50 values for this insecticide in *cyc* dsRNA larvae remained at a low level throughout the day, which was significantly lower than in the control (the maximum difference was found in Zt 8 where it was more than seven times), and which was comparable to the lowest value found for control insects at the end of night (Zt 20). In insects from the *pdp1* dsRNA group, these differences were even more significant. The largest reduction in the LC50 value, in relation to the LC50 in control, was found during the evening hours (Zt 8), when it was nearly eleven times that of the control, and the lowest at the end of the night (Zt 20), when it was approximately twice that of the control (Figure 5A). This significant reduction of average LC50, to the comparable values maintained throughout the entire day, also resulted in disappearance of the susceptibility rhythm to fipronil in larvae from both *cyc* and *pdp1* dsRNAs groups (Appendix A).

Susceptibility to deltamethrin (as expressed by changes in the LC50) in control larvae changed cyclically over 24 h with a trough in the evening (Zt 8) and a peak at the end of the night (Zt 20); the difference was more than three-fold (Figure 5B). The cosinor analysis of this oscillation showed that it had the characteristics of a rhythm (Appendix A). Interestingly, in all experimental groups of larvae treated with dsRNAs, the mean LC50 values for deltamethrin at most time points of the day were kept at a low level, and only in *tim* and *cry2* dsRNA insects (same as in control) was there a rapid increase from the middle of the night (Zt 16) to the maximum at the end of the night (Zt 20) (Figure 5B). Such an increase was not recorded in *per*, *cyc*, and *pdp1* dsRNA insects, therefore, the average LC50 values for deltamethrin from these larvae exposures were significantly lower at the end of the night (Zt 20) when compared to the control, by more than four times, seven times, and 240 times for *per*, *cyc*, and *pdpi1* dsRNA insects, respectively (Figure 5B). Furthermore, cosinor analysis showed that the treatment with *tim* and *cry2* dsRNAs did not abolish the deltamethrin susceptibility rhythm in larvae, expressed as daily changes in the average LC50 value (Appendix A). This rhythm, however, disappeared in larvae treated with *per*, *cyc*, and *pdp1* dsRNAs.

The control larvae showed clear diurnal differences in the susceptibility to malathion, as reflected in the LC50 after exposure at different times of the day. LC50 was lowest in the morning (Zt 0/24) and the highest at the beginning of the night (Zt 12), and the difference was more than three-fold (Figure 5C). Moreover, the cosinor analysis of these data proved that the larvae exhibited a malathion susceptibility rhythm (Appendix A). Analysis of the diurnal LC50 profile for malathion showed a reduction in its values for the larvae injected with *per*, *cyc*, and *pdp1* dsRNA at each time point of the day when compared to the control values. The greatest differences were recorded at the turn of the day and night (Zt 12) with a value that was almost sixteen, eighteen, and eleven times lower than in the control for the *per*, *cyc*, and *pdp1* dsRNA-treated larvae, respectively (Figure 5C). This very significant reduction of the LC50 values from malathion exposure of *per*, *cyc*, and *pdp1* larvae at each time point was also reflected in the flattening of the diurnal profile of its changes, indicating a loss of rhythm in susceptibility to malathion of these larvae (Appendix A). The situation was completely different in the case of *tim* and *cry2* dsRNA larvae, in which such a daily profile of average LC50 values for malathion from exposed larvae was maintained, i.e., similar to that observed in the control (Figure 5C). Thus, the lowest LC50 values were noted in the morning (Zt 0/24) and the highest at the beginning of the night (Zt 12); in both *tim* and *cry2* dsRNA insects, the increase was by approximately two and a half times. In addition, cosinor analysis of the LC50 values from exposure to malathion indicated rhythmic changes over the day in susceptibility of *tim* and *cry2* dsRNA larvae to this insecticide (Appendix A).

The insect response to propoxur treatment also varied over the course of 24 h. For the control, the lowest LC50 values for this insecticide were recorded in the morning (Zt 0/24) and the highest at the beginning of the night (Zt 12), with a difference of a little more than two-fold (Figure 5D). The daily changes in the susceptibility of control larvae to propoxur expressed by changes in LC50 had the characteristics of a rhythm, which is confirmed by cosinor analysis (Appendix A). In larvae treated with *per*, *cyc*, and *pdp1* dsRNA, in addition to propoxur, the mean LC50 values for this insecticide were lower (without exception) at each time point when compared with the LC50 values for the control. and the greatest differences were noted at the beginning of the night (Zt 12), with nearly twenty-six-fold difference for *per* dsRNA, twenty-eight-fold difference for *cyc* dsRNA, and nearly nineteen-fold difference for *pdp1* dsRNA larvae (Figure 5D). While in the case of larvae treated with *cyc* and *pdp1* dsRNAs, the LC50 values from propoxur exposure remained comparatively low throughout the day. In the case of larvae treated with *per* dsRNA in the morning (Zt 0/24 and 24/0) and at the beginning of the night (Zt 12), LC50 values were lower than in other time points of the day (Figure 5D). However, no rhythmic changes were observed in the LC50 values of the insect groups (*per*, *cyc*, and *pdp1* dsRNAs) for propoxur, which would indicate diurnal differences in susceptibility to this compound (Appendix A). In larvae treated with *tim* and *cry2* dsRNAs and exposed to propoxur, the LC50 value changed within 24 h, and the profile of these changes was similar to that in control insects. However, here, the lowest LC50 values were noted in the morning (Zt 0/24) and the highest at midnight (Zt 16), with a difference of almost two-fold in both cases. In addition, cosinor analysis of the LD50 values from propoxur exposure indicated the maintenance of susceptibility rhythms in *tim* and *cry2* dsRNA larvae to propoxur (Appendix A).

Response of insects from the control group to acetamiprid varied over 24 h (Figure 5E). As expressed by the different LC50 values, such changes have the characteristics of a biological rhythm (Appendix A): with an evening peak (at Zt 8) and a late night trough (at Zt 20), with a nearly three-fold difference between LC50 at these time points. Injection of insects exposed to acetamiprid with the dsRNAs of the *per*, *tim*, and *cry2* genes resulted in a significant elevation of the LC50 value for this insecticide. The highest difference in values of LC50 for acetamiprid between the control and *per* and *tim* dsRNA insects were found at the end of the night (Zt 20), and in the morning for *cry2* dsRNA insects (Zt 24/0). These differences were more than ten-fold, eleven-fold, and nineteen-fold, respectively (Figure 5E). It should be noted, however, that the LC50 values for acetamiprid in each of these insect groups remained almost constant throughout the day and night (at each time point). This, in turn, is reflected in the lack of rhythm changes for this parameter during the 24 h, and thus, the lack of daily susceptibility of studied larvae to acetamiprid (Appendix A). Similarly, the loss of this rhythm was also observed in insects treated with acetamiprid and additionally injected with *cyc* and *pdp1* dsRNA. However, in the *cyc* dsRNA group, the LC50 values from acetamiprid exposure were comparable to or lower by up to two to five times (from Zt 4 to Zt 12) that of the control; while LC50 in the *pdp1* dsRNA group was lower at all time points than in the control (Figure 5E). This difference between values for *pdp1* and control larvae was sometimes very significant, as much as nearly ten times in the evening (Zt 8).

Analysis revealed diurnal differences in the control larvae response to imidacloprid. The highest LC50 value from imidacloprid exposure was found in the evening (Zt 8) and the lowest at the beginning of the night (Zt 12), with the difference being slightly more than two-fold (Figure 5F). These diurnal differences in LC50 values after cosinor analysis indicated that the tested larvae were characterized by a rhythm of susceptibility to imidacloprid (Appendix A). Administration of *per*, *tim,* and *cry2* dsRNA resulted in a significant increase of the LC50 value for imidacloprid-treated larvae treated for most of the day, with exceptions for *per* and *tim* larvae treated in the middle of the day (Zt 4) and in the evening (Zt 8), for which LC50 for imidacloprid was similar to that of the control. The highest (about three times) increase of LC50 after the injection of *per* and *tim* dsRNA in relation to the control was found at the beginning of the night (Zt 12) and after *cry2* dsRNA injection (more than five times) in the morning (Zt 0/24) (Figure 5F). No cyclic changes in the LC50 for imidacloprid were found in any of these groups of larvae, indicating a disappearance of the rhythm to imidacloprid susceptibility following such treatment (Appendix A). The complete disappearance of this rhythm was also found in the larvae given *cyc* and *pdp1* dsRNA. However, injection of these dsRNAs resulted in a significant increase in susceptibility to imidacloprid as the LC50 values were significantly lowered in these groups of larvae compared to the control (Figure 5F). The maximum difference in this comparison for *cyc* and *pdp1* dsRNA larvae was recorded in the evening (Zt 8) by more than four-fold and twenty-six-fold, respectively.

Drawing general conclusions, our observations indicated a clear influence of the molecular oscillator of *S. littoralis* larvae on their susceptibility to insecticides. When looking for the TTFL components of the oscillator that are particularly important in response to insecticides, *pdp1* and *cyc* are noteworthy. The knockdown of the expression of these two genes resulted in a radical increase in the susceptibility of larvae to all tested insecticides. Such a strong effect in response to all insecticides was not found when the expression of the *per* gene is lost. However, it was evident with fipronil, deltamethrin, malathion, and propoxur. An analogous relationship need not necessarily be the case for the rest of the oscillator genes when they are knocked down. In some cases, as indicated by the studies shown above, the tendency was even opposite—disturbance in the expression of clock genes *per*, *cry2*, and *tim* led to a very significant decrease in insect susceptibility to selected insecticides—such a relationship was evident in the case of neonicotinoids (acetamiprid and imidacloprid). However, even if susceptibility to selected insecticides decreased after knockdown of clock genes, insects treated in this way have a disturbed rhythm of response to these compounds. The results indicated that the abolition of the smooth operation of the oscillator in *S. littoralis* larvae may significantly impair their metabolic response to xenobiotics.

## 4. Discussion

The realization of the vital functions of each organism at the right time of the day is the key to the efficient course of physiological processes maintaining homeostasis. Such processes are aimed at optimizing the use of energy necessary for steady state maintenance, and are accomplished by an efficiently operating biological clock mechanism. Among the numerous biological clock-dependent processes, those related to the response to compounds that are not physiologically present in the organism and delivered from the environment (xenobiotics) have a particularly high adaptive value. It has been proven that the disturbance of the clock functions has an extremely destructive effect on the body through inefficient metabolism and excretion of xenobiotics, which, when remaining in a biologically active form, threaten the homeostasis of many organs and body systems. The results presented here are an example of the role of the biological clock in minimizing the negative impact of xenobiotics on the organism. We found that *S. littoralis* larvae are less susceptible to insecticides if their biological clock is functioning properly. On the other hand, if we disrupted the larvae’s clock by exposure to constant light (numerous rhythmic processes in moths are abolished under these conditions [59]), their susceptibility to most frequently used insecticides increased. The difference in susceptibility of larvae from groups kept in LL and LD was greatest at the time when the LC50 for LD larvae was highest. This was obvious when we considered that the susceptibility of larvae from LD conditions to insecticides (expressed by changes in the LC50 value) fluctuated in the daily rhythm, while in insects from LL conditions, it remained at a comparable level throughout the entire day. In the case of the three insecticides tested—deltamethrin (pyrethroid), malathion (organophosphate), and propoxur (carbamate)—the LC50 values were significantly higher for LD larvae at night than during the light phase of the day; while the LC50 for fipronil (phenylpyrazole), acetamiprid, and imidacloprid (both neonicotynoids) applied to the LD larvae had higher values during the day and lower values at night. Moreover, in the case of imidacloprid, at two time points of the night, the LC50 value was higher for insects reared in LL than in LD. This suggests that clock dysfunction not only disrupts the daily response to insecticides but may also contribute to increased resistance to some of these compounds. The circadian susceptibility and resistance of insects to various insecticides, observed at the level of the individual or population, was the subject of several reports. Daily oscillations in sensitivity to different poisons have been reported in cockroaches, beetles, flies, and moth species, with particular attention to the insecticides commonly used in the 1960s, such as DDT, dieldrin and endrin (organochlorides), malathion, dichlorvos, parathion and azinphosmethyl (organophosphates), dimetilan (carbamate), and pyrethrum extracted from plants [60]. Analysis revealed circadian rhythms in susceptibility and resistance to dieldrin of the migratory locust, *Locusta migratoria migratorioides* larvae that were more susceptible to this insecticide when dosed during the night [61]. Similarly, as in our studies, daily susceptibility rhythm to propoxur was abolished in *D. melanogaster* reared under constant light (LL) [45]. However, the daily rhythm of susceptibility to this insecticide had a different profile in *D. melanogaster*. Based on LC50 analyses, *D. melanogaster* had the lowest susceptibility to propoxur during the day, whereas susceptibility was lowest for *S. littoralis* at night.

Over the last decade, studies on the daily changes in susceptibility to insecticides continue to be relevant. Oscillations in sensitivity to esfenvalerate, cyhalothrin, alpha-cypermethrin, beta-cyfluthrin, deltamethrin (pyrethroids), pyriproxyfen (derivative of pyridine), pirimicarb (carbamate), diazinon (organophosphate), indoxacarb (organochlorine), and teflubenzuron (benzoyl urea derivative) were shown in the honey bee *Apis mellifera* [62,63]. In this case, the percentage of survivors changed over the day, depending on the time when they were intoxicated and type of insecticide used. This, in turn, resembles the relationships we have observed for *S. littoralis*, where daily profiles of larvae susceptibility to individual insecticides varied greatly throughout the day. Interestingly, susceptibility to diazinon (organophosphate) in cotton aphid *Aphis gossypii* was shown as changing during the day with a fashion similar to what we observed in *S. littoralis* treated with malathion (also organophosphate) [64]. Moreover, exposure to continuous lighting also resulted in the disappearance of the cyclical changes in LC50 values for diazinon in aphids, as is the case with LC50 for malathion in *S. littoralis*. The response to pyrethoids seemed to be similar in *S littoralis* and in another moth, *B. mori*, to deltamethrin and permethrin, respectively [65]. As we showed for *S. littoralis*, *B. mori* also had the lowest susceptibility to these insecticides in the second half of the night, and the highest during the day. Crickets, *Acheta domesticus*, also showed the highest resistance to the pyrethroid ß-cyfluthrin at night, expressed as a high survival rate [66]. The scope of research mentioned above was significantly expanded when the presented results also concerned the analysis of specific gene expression. Genes encoding the enzymes for xenobiotic metabolism in general (and insecticides in particular) with an emphasis on the families of Glutathione-S-transferases (GSTs) and cytochrome P450 monooxygenases (P450s) have been described as rhythmically expressed in the yellow fever mosquito *Aedes aegypti* [67]. Such expression properties seem to be related to the diurnal fluctuations in different xenobiotics sensitivity of this species [68,69]. In this regard, more attention was paid to the malaria vector mosquito *Anopheles gambiae*. Cyclic changes of numerous genes expression which are involved in the metabolism of DDT and pyrethroids were shown, and both were determined at the level of the transcripts [70,71] and at the level of the selected proteins [72]. Interesting results were obtained by mimicking the field conditions in the laboratory and showing the diurnal variability of the response of the *Drosophila suzukii* population to malathion (organophosphate), and lack of such a response rhythm to fenpropathrin (pyrethroid), while indicating that the expression of a number of genes for detoxification factors is rhythmic in this species [73]. In the common bed bug, *Cimex lectularius*, susceptibility to pyrethroids and neonicotynoids was also studied in the context of the circadian regulation of genes for GSTs and P450s, linking observations with the daily activity of these enzymes [74]. Variable transcription of genes for nicotinamide adenine dinucleotide phosphate cytochrome P450 reductase (CPR) and selected P450s involved in the development of pyrethroid resistance were studied in the kissing bug, *Triatoma infestans*, a vector of *Trypanosoma cruzi* causing Chagas disease [75]. However, rhythmic changes in gene expression for enzymes related to detoxification in studied organs did not necessarily correspond to daily changes in the amount and the activity of these enzymes. Unfortunately, the number of data verifying whether such rhythms occur in the organs of insects is very sparse. Likewise, there are only a few reports proving the relationship between the TTFL elements of the oscillator and its influence on the activity of detoxifying enzymes. The reasons include the lack of mutants of the biological clock genes in most species or the rare use of efficient molecular tools whose application would knock out expression of these genes. In *S. littoralis*, the application of the RNAi technique with the use of dsRNA turns out to be effective [37,50]. Therefore, the administration of dsRNA to insects generated on the fragments encoding clock genes effectively knocks out their expression in organs, including those heavily involved in detoxification, such as, the fat body, midgut, and Malpighian tubules. We chose four genes from the main TTFL (*per*, *tim*, *cry2*, and *cyc*) and one so-called supporting TTFL (*pdp1*); *cyc* and *pdp1* occupy a special position, as both encode factors involved in the regulation of cellular responses to xenobiotics. The use of dsRNA not only destroyed the rhythmic expression of the clock genes, but also very significantly lowered the level of transcripts in the studied organs. This effect on clock genes expression has an influence on the daily profiles of enzymatic activity related to detoxification (P450, GST, and esterase) in all examined organs. Nevertheless, this effect very much depended on which gene expression was influenced and which organ was tested. We can conclude that dysfunction of *per* expression lowers the activity of P450 and increases GST and esterase in all examined organs. Disturbance of *tim* expression, in principle, has little or no effect on the function of P450 and GST, but essentially increases esterase activity in the fat body and midgut. On the other hand, knockout of *cry2* expression does not change P450 activity in the fat body and Malpighian tubules, and increases it in the midgut. In the same larvae, an increase is also observed in the GST and esterase activity in the midgut and Malpighian tubules. We observed a very homogeneous effect after administering the *cyc* and *pdp1* dsRNA to insects. Without exception, this led to a decrease in the activity level of each of the tested enzymes in each of the examined organs. Moreover, despite the knockout of *per*, *tim*, and *cry2* expression, in some organs, the rhythmic activity of the tested enzymes was maintained, especially after injection of *tim* dsRNA into larvae. However, the enzymatic activities tested were not found to have signs of biological rhythm in any of the organs when the expression of *cyc* and *pdp1* was disturbed. This may be explained by the high activity of enzymes around the clock in insects that were injected with dsRNA of genes encoding *per* and *cry2* due to subordination of genes encoding these enzymes to PER and CRY2 proteins. This relationship would have the form of negative regulation. Thus, the lack of factors encoded by *per* and *cry2* enables the expression of detoxification genes during this phase of the day when its level normally remains low. On the other hand, the persistently low level of enzyme activity in *cyc* and *pdp1* larvae indicates that both TTFL elements considered as positive regulators of transcription are necessary to maintain gene expression for detoxification enzymes at an elevated level at certain times of the day. The clearly less significant effect of the *tim* dsRNA on the enzymatic activities studied proves the minor importance of this gene in the regulation of enzymes involved in detoxification, or more generally, in the metabolism of xenobiotics. This might not be so surprising considering the role of *tim* in the oscillator itself in lepidopterans, which is rather supporting and related to the information about photoperiodic status of the environment than regulation of genes expression that are dependent on the clock. To date, rhythm analysis of GST activity, esterase, and oxidase activity in the heads and whole bodies of *A. gambiae* indicated that only GST activity had features that allowed it to be considered rhythmic in both parts of the insect body, with a distinct peak extending for hours from late night to dawn [71]. In turn, esterase and oxidase activity are not rhythmic in this species, which the authors explain primarily as a consequence of the method used, i.e., work on the entire body parts, which contain many tissues and organs with different physiological parameters and with differently ticking peripheral oscillators. Similarly, in the abdomen of *D. melanogaster*, esterase activity, and GST were found to be non-rhythmic, but ECOD (P450) and uridine 5′-diphosphoglucosyltransferase (UGT) activities were rhythmic, with a peak in the first half of the day and the second half of the night, respectively [45]. In *C. lectularius*, the activity of ECOD (P450), GST, and esterase were clearly rhythmic with peaks in the evening hours, although the measurements of activity were made in protein extracts obtained from whole insects [74]. Although we did not perform analyses on whole insect extracts, it appears that the activity measured in this way would be burdened with many artifacts, as the daily activity profiles of each enzyme in different organs were not identical, and while the activity of GST in the fat body, midgut, and Malpighian tubules in our larvae were somewhat similar, the activity of ECOD in the fat body and Malpighian tubules differed from that in the middle intestine, and the activity of esterases in Malpighian tubes was completely different than in the fat body and intestine. In the closely related species, *S. litura*, GST activity in the hemolymph was significantly lower when the clock was desynchronized by growing insects under constant light [76]. Additionally, whole body GST and esterase activity in the five *A. aegypti* strains was significantly higher when the insects were kept under the optimal photoperiod, and lower when transferred to permanent darkness, confirming the significant role of the oscillator in the body’s response to xenobiotics [77]. Our considerations show that the new path in research may be marked by the possibility of using the RNAi method, where the target will be elements of systems regulating enzymes involved in the metabolism of xenobiotics, and whose activity is naturally rhythmic. It seems that disturbing the expression of the biological clock genes that code for regulatory factors that rank high in the regulation of xenobiotic responses could be an excellent tool. The RNAi used to control susceptibility to insecticides has the advantage that switching off the expression of individual components will lead to the dysfunction of many subordinate metabolic pathways. Thus, the action is systemic and more complex than when the targets are single genes encoding detoxification enzymes, often postulated as key switch-off targets using the RNAi technique [78]. Among them are those that encode enzymes characterized by the activity we are studying, i.e., P450, GST, and esterase. This reasoning was also followed by the authors of the paper, showing the possibility of switching off the expression of detoxifying genes (however, the activity of enzymes was not studied) in *S. litura* treated with the dsRNA of clock genes [20].

The interest in using RNAi to protect crops by controlling insect pests is high and increasing, as reflected in the literature. The proposed target in this case are genes usually related to both development and metabolism in general [79,80,81,82]. Among the latter, knockout of these involved in the inactivation and removal of xenobiotics from the body is of special importance, as they are shaping the susceptibility of insects to insecticides [83,84,85,86]. The use of RNAi to increase the susceptibility of insects to insecticides is also logical and used, for example, in strains that have developed resistance to these compounds [87,88,89]. RNAi is also applied in research on the development of resistance [90].

In our research, we showed that disrupting the function of a molecular oscillator by using RNAi to knock out clock genes (and therefore encoding factors that affect transcription) effectively changed the activity of detoxifying enzymes in dedicated organs (as it is mentioned above), but also affected the susceptibility of *S. littoralis* larvae to the insecticides we have tested. Several conclusions can be drawn from our results. First, the treatment of insects with *per* and *cyc* (almost at every point of the day) and *pdp1* (at every time point of the day) significantly lowered the LC50 for fipronil, malathion, and propoxur. Second, the susceptibility of insects (LC50) exposed to these insecticides did not change substantially over the course of the day following the injection of *tim* and *cry2* dsRNA. Third, the effect of dsRNA treatment in insects exposed to deltamethrin was primarily observed almost exclusively during the hours when the LC50 value was highest in the control group (i.e., at night). An exception is the administration of *pdp1* dsRNA, which dramatically lowered the LC50 for deltamethrin, regardless of its administration time. Fourth, the reaction of larvae to neonicotinoids after treatment with dsRNA of genes encoding negative regulators in TTFL (*per*, *tim*, and *cry2*) was similar, i.e., the LC50 value was increased compared to that of the control larvae. However, when the dsRNA of positive regulators in TTFL (*cyc* and *pdp1*) were used, the LC50 value for these insecticides decreased or (less frequently and only in *cyc* dsRNA larvae) did not change. 

Biological clock genes represent a specific class as, when encoding transcriptional regulatory factors, they are highly situated in the cascade of the organism’s response to noxious substances. Using the RNAi gene silencing technique, the significant role of genes that act in a such hierarchical system has also been demonstrated, which, by coding for factors regulating transcription are superior in the control of executive genes during detoxification [85,91]. However, as we mentioned above, data on the relationship between silencing the expression of clock genes and changes in the physiological response, e.g., in the form of differences in susceptibility to insecticides, are only few and come from the study of *D. melanogaster* mutants, and after RNAi treatment in *S. litura* and *A. aegypti* [18,45,46]. This issue seems to be very interesting and future-oriented, because it draws our attention to the possibility of a rational use of insecticides, mainly by applying different doses depending on the time of day. On the other hand, it will minimize the scale in which they can be used. In addition, the use of various methods that result in silencing gene expression, especially those highly located in the hierarchy of homeostasis control, gives the possibility of an effective and species-targeted tool for controlling the number of their populations with no or negligible impact on other organisms. A reflection of the interest in these issues are the numerous studies of the so-called RNAi insecticides that we can read about in a rapidly growing number of scientific papers [80,92,93]. The summary of our research may raise the question of which of the TTFL elements in the *S. littoralis* larvae oscillator are particularly involved in the response to insecticides. The answer to this question is not easy, because the knockout of different genes resulted in different reactions of the larvae. However, attention may be drawn to the fact that both the susceptibility to insecticides and the activity level of the tested enzymes were decreased in each of the tested organs when *pdp1* expression was disturbed; the same is true in most cases when insects were treated with *cyc* dsRNA. These observations suggest a very strong involvement of the second TTFL oscillator in the response to insecticides, and its dysfunction results in a decrease in insecticide resistance of the larvae. Such a suggestion is consistent with the postulate put forward by the authors of the work on *D. melanogaster* [18]. Lowering the level of the available PDP1 factor, either directly by affecting its gene or indirectly by reducing the availability of the *pdp1* positive transcriptional regulators (i.e., *cyc*) produced a similar effect. Such indirect regulation dependent on factors positively regulating transcription in the TTFL seems to be a case described in *S. litura*, however, the RNAi knockout was performed on the *cwo* gene [20]. It has been shown that lowering the expression of *cwo* caused an increase in the expression of genes involved in detoxification, because CWO likely competes with clock transcription factors (CLK-CYC) at the binding site in regulatory regions of these genes. The undoubted importance of research in this area is also emphasized by the results of analyses conducted on mammals, indicating that the CLK and BMAL1 (mammalian CYC homolog) loop is an important point in the path of xenobiotic metabolism, coordinating efficient detoxification [94,95,96]. Some dissonance with commonly accepted knowledge may cause a discrepancy in the body’s physiological response to the dsRNA of the *per* and *cry2* genes—on the level of detoxification enzyme activity and insecticide susceptibility that we tested. The PER and CRY2 proteins function as a heterodimer in the TTFL loop of the oscillator. One would therefore expect a similar or even identical effect to be induced by *per* and *cry2* dsRNA administrations. However, it must be taken into account that the organism’s response to the knock down of each of these genes does not have to manifest only in a change in the functioning of the molecular oscillator and the genes dependent on the oscillator. The PER and CRY2 proteins may also perform other functions in the body, which is the subject of numerous discussions. Especially the one concerning peripheral oscillators—e.g., the involvement of oscillator genes, especially *per*, in response to genotoxic stress [97]. Perhaps we could also deal with such a function of *per* and *cry2* in the response to stress induced by xenobiotics. Undoubtedly, it is an excellent problem for planning research in the near future.

## 5. Conclusions

The conducted experiments led us to the conclusion that molecular oscillators of lepidopterans larvae (most likely active locally in peripheral tissues) are responsible for the control of the xenobiotic metabolizing enzymes in organs strongly involved in detoxification, such as the fat body, midgut and Malpighian tubules. The activity of these enzymes, which changes throughout the day, is most likely responsible for the daily cycle of susceptibility of larvae to insecticides. Ultimately, we postulate that these results support the idea of the rational use of insecticides at strictly defined times of the day.

## Data Availability

On reasonable request, derived data supporting the findings of this study are available from the corresponding author.

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
