# Peer review of "Molecular Oscillator Affects Susceptibility of Caterpillars to Insecticides: Studies on the Egyptian Cotton Leaf Worm—Spodoptera littoralis (Lepidoptera: Noctuidae)"

_insects, 2022, doi:10.3390/insects13050488_

Round 1

Reviewer 1 Report

The paper titled “Molecular oscillator affects susceptibility of caterpillars to insecticides: Studies on the Egyptian cotton leaf worm – Spodoptera littoralis (Lepidoptera: Noctuidae)”, it is of great interest both for the knowledge of the general biology of S. littoralis, as well as for the area of use of insecticides against this organism. However, it has some aspects that must be corrected and taken into account.

Method:

Nowhere in the method is it mentioned which bioethical guidelines for the treatment and handling of animals were followed to carry out the experimental design and handling of animals, this must be included.

P3, L147

How many larvae in total were used for the entire experiment?. Put the n=??

P4, L190-191

Did the authors homogenize the tissues of each organ with a buffer, or did they store them intact and dry?. If they were homogenized, did they use the pellet, the supernatant, or all of it?. Specify this.

P25, L1078

“Drosophila”, it was missing to write in cursive letters.

Author Response

We would like to thank the reviewer for providing suggestions that have a positive impact on the quality of our manuscript. Below we have answered the questions and information on how we solved the problem.

Yours sincerely,

Piotr Bębas

Nowhere in the method is it mentioned which bioethical guidelines for the treatment and handling of animals were followed to carry out the experimental design and handling of animals, this must be included.

Our response to the reviewer:

The Polish guidelines do not require any special procedures for dealing with insects in scientific experiments, except for protected species; S. littoralis is not included in this group. Nevertheless, all tests were performed respecting the animals, and the number of animals used for the analysis was reduced to the necessary minimum. Before all procedures, the larvae were anesthetized in a CO2 atmosphere; in addition, sections were performed in ice-cold buffer. The text of the manuscript has been supplemented with the information on the procedure for dealing with insects. 

P3, L147

How many larvae in total were used for the entire experiment?. Put the n=??

Our response to the reviewer:

We would like to slightly argue with the reviewer. We give in the text how many individuals were used in a given group (range from 10 to 15 and sometimes from 10 to 12) and how many repetitions we did. For each analysis, we used a total of approximately 5,000 to 5,500 individuals. It seems to us that posting such information is not fully justified if we are writing in the manuscript about the number of repetitions and the number of individuals in the group.

P4, L190-191

Did the authors homogenize the tissues of each organ with a buffer, or did they store them intact and dry?. If they were homogenized, did they use the pellet, the supernatant, or all of it?. Specify this.

Our response to the reviewer:

Following the reviewer's suggestion, we clarified the information on the collection of organs and the initial preparation of the material for extraction. We direct the reader to the following paragraphs, because it is indicated there whether we used pellets or supernatants.

P25, L1078

“Drosophila”, it was missing to write in cursive letters.

Our response to the reviewer:

Thank you for checking the text in detail. Of course, we improved this entry, the font form of the genus name.

Reviewer 2 Report

Circadian clocks control the rhythmicity of many behaviors and physiological features of insects. Here, M. Haj Darwich and colleagues showed susceptibility of Spodoptera littoralis larvae to insecticides varies daily, being under the control 24 of a molecular oscillator. Moreover, they found that the oscillator operates in the fat body, midgut, and Malpighian tubules where it is likely to control the activity of detoxifying enzymes. These results strongly indicate the role of a molecular oscillator in the metabolism of xenobiotics in S. littoralis larvae.

I have the following concerns:

  1. Does the amount of consumed food show rhythmic changes for Spodoptera littoralis larvae under LD conditions?

  1. During the preparation of the present manuscript, data was published on the influence of a molecular oscillator on the expression of genes encoding digestion and detoxification enzymes in the lepidopteran representative, Spodoptera litura. Here, the authors refer to Ref.39. I think they should refer to Ref.20. In the Ref.39, it is talking about the circadian rhythm system of zebrafish (Danio rerio).

  1. The insecticide concentrations used were as follows: 0.5–250 μg/mL fipronil, 0.1–100 μg/mL deltamethrin, 0.5–120 μg/mL malathion, 0.1–164 μg/mL propoxur, 0.1–350 μg/mL acetamiprid, and 0.1–500 μg/mL imidacloprid. How to determine the insecticide concentrations should be stated. Moreover, the exact concentration of each insecticide should be indicated, not a range.

  1. The reference for cosinor analysis should be added and analysis methods should be mentioned in the methods. In addition, I suggest show the raw data of LC50 in Table S2.

  1. In Figure1, I suggest add a simple frame diagram to illustrate the process of insecticides treatments.

  1. In Figure2 and Figure3, I suggest also mark different enzymes name on the figures.

  1. In Lepidoptera, PER ad CRY2 are seem to work together to inhibit CLK:CYC heterodimer-dependent transcription. However, the changes of the daily patterns in the activity of xenobiotic metabolizing enzymes in the fat body, midgut, and Malpighian tubules caused by PER and CRY2 RNAi seems differently. This should be discussed. In addition, the daily susceptibility patterns for some insecticides also show difference in PER and CRY2 RNAi groups.

  1. In part 3.2, for the detoxification enzymes activity in the fat body, midgut, and Malpighian tubules, it is necessary to make a comparative summary of the rhythmic changes of enzyme activity in these three tissues

  1. Line 690-692: As a result, the cosinor analysis showed that the changes of susceptibility to fipronil in control insects had signs of a rhythm (Table S5). I think this should refer to Table S6 but not Table S5.

  1. In part 3.3, it is also necessary to make a comparative summary of the susceptibility to insecticides for different insecticides after knockdown of clock components.

  1. The second paragraph of discussion part is too long

Author Response

We would like to thank the reviewer for providing suggestions that have a positive impact on the quality of our manuscript. Below we have answered the questions and information on how we solved the problem.

Yours sincerely,

Piotr Bębas

1. Does the amount of consumed food show rhythmic changes for Spodoptera littoralis larvae under LD conditions?

Our response to the reviewer:

The answer is definitely, yes. We showed in another study that S. littoralis larvae exhibit daily oscillations in feeding rate https://doi.org/10.1016/j.jinsphys.2017.07.009 . In this case, however, it is very difficult to determine comparable oscillations in the following days. This is because S. littoralis larvae, especially in their last stages, grow rapidly. So, in the following days they eat more and more. This, in turn, has a major influence on the amplitude of the oscillations in consecutive days. Thus, considering the oscillations of one day, it can be described as a rhythm, or even as a rhythm with the parameters of the circadian rhythm. But if we combine quantitative data (without extrapolation to relative values), the mathematical analysis of the food intake rhythm is difficult - the statistical significance between the values at the same time points of the day has little statistical power.

2. During the preparation of the present manuscript, data was published on the influence of a molecular oscillator on the expression of genes encoding digestion and detoxification enzymes in the lepidopteran representative, Spodoptera litura. Here, the authors refer to Ref.39. I think they should refer to Ref.20. In the Ref.39, it is talking about the circadian rhythm system of zebrafish (Danio rerio).

Our response to the reviewer: 

Of course. Thank you very much for finding this mistake. We've already corrected it.

3. The insecticide concentrations used were as follows: 0.5–250 μg/mL fipronil, 0.1–100 μg/mL deltamethrin, 0.5–120 μg/mL malathion, 0.1–164 μg/mL propoxur, 0.1–350 μg/mL acetamiprid, and 0.1–500 μg/mL imidacloprid. How to determine the insecticide concentrations should be stated. Moreover, the exact concentration of each insecticide should be indicated, not a range.

Our response to the reviewer:

 In order to determine the LC value, we had to treat the insects with many concentrations of compounds - using them in a specific (given) concentration range, which were the same at each time point of the day. As a result, we were able to present the average LC value for a given insecticide at a given point in time on the charts. This was also the case in insects treated with dsRNA. The bars in the graphs (their size - height) indicate the value of the LC coefficient (insecticide concentration) at the selected time point. The concentration of pesticide that was lethal to 50% of flies (LC50) was calculated using PROBIT analysis i.e. a commonly used and recognizable method. The LC50 values are therefore the results - they are not the value we are setting during the experimental procedure. Therefore, it seems to us that presenting the data in the form of bars on the charts is a better solution than giving only numerical data.

4. The reference for cosinor analysis should be added and analysis methods should be mentioned in the methods. In addition, I suggest show the raw data of LC50 in Table S2.

Our response to the reviewer:

We suggest to leave the description of the cosinor method in the caption to Table S2, and to introduce a clear reference to this description into the main text. As the reviewer rightly pointed out, such information should be available. This will prevent lengthening the text of the manuscript. And at the same time, the description will be located exactly in the place (supplementary materials) where the tables are presented. There we will also introduce references to the literature on the cosinor method, on which we relied on the calculations.

We have reviewed the raw data, as suggested by the reviewer, and it seems to us that presenting it can introduce a little chaos - the data cannot be read directly. Because the LC50 calculations for correct interpretation must be subjected to PROBIT analysis, then statistical testing with the appropriate analysis of variance test and multiple comparison test (which we use to determine possible differences between values ​​at time points) and finally subjected to an algorithm that allows the calculation of cosinors. The data itself from cosinore analyzes is in a sense raw data, although already very processed for clarity of reading.

5. In Figure1, I suggest add a simple frame diagram to illustrate the process of insecticides treatments.

Our response to the reviewer:

Of course, we agree with the reviewer and have prepared a graphical diagram (Scheme I) of the procedure that we used during the experiments. As we have included Scheme I to the manuscript, we would like to replace the graphical abstract with a new one (included at the end of revised manuscript). The new graphical abstract is more general and less like Scheme I.

6. In Figure2 and Figure3, I suggest also mark different enzymes name on the figures.

Our response to the reviewer:

Of course, we agree with the reviewer and we made corrections in Figures 2, 3 and 4 (all show data obtained from enzyme activity analysis).

7. In Lepidoptera, PER ad CRY2 are seem to work together to inhibit CLK:CYC heterodimer-dependent transcription. However, the changes of the daily patterns in the activity of xenobiotic metabolizing enzymes in the fat body, midgut, and Malpighian tubules caused by PER and CRY2 RNAi seems differently. This should be discussed. In addition, the daily susceptibility patterns for some insecticides also show difference in PER and CRY2 RNAi groups.

Our response to the reviewer:

This issue is difficult to discuss. In our opinion, this is how the effect of PER and possibly CRY2 is revealed, which is not directly related to the operation of the molecular oscillator. The TTFL oscillator elements are mentioned in the literature as important regulators of other body responses as well. For example, in response to genotoxic stress. We have added the last, short paragraph of in the discussion on this topic.

8. In part 3.2, for the detoxification enzymes activity in the fat body, midgut, and Malpighian tubules, it is necessary to make a comparative summary of the rhythmic changes of enzyme activity in these three tissues

Our response to the reviewer:

We took the reviewer's suggestions into account and presented a several-sentence summary of this chapter.

9. Line 690-692: As a result, the cosinor analysis showed that the changes of susceptibility to fipronil in control insects had signs of a rhythm (Table S5). I think this should refer to Table S6 but not Table S5.

Our response to the reviewer:

Thank you very much for your in-depth analysis of the text. In fact, at this point we have mistaken the markings of the tables - the description is already corrected.

10. In part 3.3, it is also necessary to make a comparative summary of the susceptibility to insecticides for different insecticides after knockdown of clock components.

Our response to the reviewer:

We took the reviewer's suggestions into account and presented a several-sentence summary of this chapter.

11. The second paragraph of discussion part is too long

Our response to the reviewer:

We would be very grateful to the reviewer for accepting the length of the discussion as it is now. We realize that it is extensive in several areas and possibly too long. However, it seems to us that the problem of detoxification of xenobiotics is very hot in the context of regulation by the biological clock. Especially when looking at the problem from the perspective related to the possibility of using modern tools of molecular biology in research. Extensive discussion allows us to refer also to numerous literature items. We hope that the discussed issue will be of real interest to a wide audience.

Reviewer 3 Report

The results indicate the role of a molecular oscillator in the metabolism of xenobiotics in S. littoralis larvae and showed that the susceptibility of S. littoralis larvae to insecticides varies daily, being under the control of a molecular oscillator. This oscillator operates in the fat body, midgut, and Malpighian tubules where it is likely to control the activity of detoxifying enzymes. The findings in the present may indicate clock genes as potential targets of molecular manipulation to produce plant protection compounds based on the RNAi method. The study is significant.

The following contents are should be improved.

L160:“No longer than 2 h” what is the exact time? 1h? 0.5h? or 1.9h? or others?

Results:The results should be shorted and cannot repeat the methods or the aim or the background. It will be better if only the results should be shown in the section and the other description (such as L270-281, etc.) should be move to the other places such as “2. Material and methods” or “4. discussion”. In addition, some contents have been descripted in the material and methods, and should not to be repeated or should be deleted in the results such as L281-289, etc.

Discussion: in the part, suggest to decrease the results which were descripted too much, and the part should be shorted.

Author Response

We would like to thank the reviewer for providing suggestions that have a positive impact on the quality of our manuscript. Below we have answered the questions and information on how we solved the problem.

Yours sincerely,

Piotr Bębas

L160“No longer than 2 h” what is the exact time? 1h? 0.5h? or 1.9h? or others?

Our response to the reviewer:

We thank the reviewer for this remark. The thought was poorly worded. We started the vial coating procedure 2 hours before the insects were exposed. But sometimes the coating procedure took a little less time. Nevertheless, the insects were always placed in the vials at the appropriate point in time. We corrected this ambiguity in the manuscript.

ResultsThe results should be shorted and cannot repeat the methods or the aim or the background. It will be better if only the results should be shown in the section and the other description (such as L270-281, etc.) should be move to the other places such as “2. Material and methods” or “4. discussion”. In addition, some contents have been descripted in the material and methods, and should not to be repeated or should be deleted in the results such as L281-289, etc.

Our response to the reviewer:

Thank you for these comments. We took them into account and made changes to the text. We hope it is exactly as the reviewer suggested. Unfortunately, in the revision manuscript, the line numbering shifted slightly. But to follow the idea and intentions of the reviewer (we hope we understood them well) we made corrections as best we could. We have removed the repetitions of methods and discussions from the Results section.

Discussion: in the part, suggest to decrease the results which were descripted too much, and the part should be shorted.

Our response to the reviewer:

Thank you for this suggestion. We recognize that the discussion in the manuscript is very extensive. The chapter is long, and maybe actually reading it can be tedious. But our intention was to collect as much literature as possible on the relationship between the oscillator (mainly the molecular one) and the detoxification of insecticides in various species. We wanted to discuss our results against the background of this literature as thoroughly as possible. Recently, there is more and more work in this area and the topic is getting hot. When we consider the possibility of using the RNAi method to control insects, the clock genes seem to be an excellent tool. Therefore, we would like to ask the reviewer to accept such an extensive discussion.